# Electrostatic Forces in Control of the Foamability of Nonionic Surfactant

Stoyan I. Karakashev [1,*], Nikolay A. Grozev [1], Svetlana Hristova [2], Kristina Mircheva [1] and Orhan Ozdemir [3]

1   Department of Physical Chemistry, Sofia University, 1 James Bourchier Blvd, 1164 Sofia, Bulgaria
2   Department of Medical Physics and Biophysics, Medical Faculty, Medical University–Sofia, Zdrave Str. 2, 1431 Sofia, Bulgaria
3   Department of Mining Engineering, Istanbul University-Cerrahpaşa, Buyukcekmece, Istanbul 34320, Turkey
*   Correspondence: fhsk@chem.uni-sofia.bg

**Abstract:** Can the DLVO theory predict the foamability of flotation frothers as MIBC (methyl isobutyl carbinol)? The flotation froth is a multi-bubble system, in which the bubbles collide, thus either coalescing or rebounding. This scenario is driven by the hydrodynamic push force, pressing the bubbles towards each other, the electrostatic and van der Waals forces between the bubbles, and the occurrence of the precipitation of the dissolved air between the bubbles. We studied the foamability of 20 ppm MIBC at constant ionic strength I = $7.5 \times 10^{-4}$ mol/L at different pH values in the absence and presence of modified silica particles, which were positively charged, thus covering the negatively charged bubbles. Hence, we observed an increase in the foamability with the increase in the pH value until pH = 8.3, beyond which it decreased. The electrostatic repulsion between the bubbles increased with the increase in the pH value, which caused the electrostatic stabilization of the froth and subsequently an increase in the foamability. The presence of the particles covering the bubbles boosted the foamability also due to the steric repulsion between the bubbles. The decrease in the foamability at pH > 8.3 can be explained by the fact that, under such conditions, the solubility of carbon dioxide vanished, thus making the aqueous solution supersaturated with carbon dioxide. This caused the precipitation of the latter and the emergence of microbubbles, which usually make the bubbles coalesce. Of course, our explanation remains a hypothesis.

**Keywords:** frothers; foamability; DLVO; interfacial forces; fine particles; zeta potential

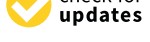



## 1. Introduction

The production of foam/froth is a complex process of the formation of myriads of bubbles in an aqueous medium, thus resulting in the froth observed on the top of the liquid [1–3]. There are many works that study the correlation of the foam stability and/or the foamability of surfactant aqueous solutions (e.g., [4–30]) with, for example, the foam lamellae elastic moduli/Gibbs elasticity [3–5,8,11,25], the state of the surfactant adsorption layer [1,6,11,26,31,32], the presence of particles [1,6,7,23,27–29], the behavior of foam films [1,12,30,32–38], the existence of special stimuli-responsive reagents [39–43], etc., but probably due to the complexity of the object, none of them sought to investigate the correlation between the electrostatic repulsion between the bubbles and the foamability of froths. Yet, the electrostatic stabilization of hydrophobic dispersions is the first principle of stabilization according to the celebrated DLVO theory [44–48]. For this reason, the thin film pressure balance method (TFPB) has been applied to study the dependence of the disjoining pressure between film surfaces on the thickness of the foam films [49–52]. Therefore, foam (emulsion) films with stronger electrostatic repulsion between film surfaces should correspond to more stable foams (or emulsion). Unfortunately, the opposite trend has been established for the case of foams (emulsions) stabilized by ionic surfactants in the presence of different added electrolytes [52]. Our approach is holistic and inductive. For this reason, we prefer to study a simpler system, i.e., the effect of the electrostatic repulsion

between the bubbles on the foamability of an aqueous solution of nonionic frother (e.g., MIBC). The intrinsic negative surface potential of the bubbles in water at normal pH value is still not well understood, but it is proved that it depends on the pH value [53]. Their isoelectric point is at pH ≈ 4 [1,54]. Hence, we decided to perform a simplified experiment, in which we controlled the zeta potential of the bubbles by varying the pH values of an aqueous medium at constant ionic strength I = $7.5 \times 10^{-4}$ mol/L in 20 ppm methyl isobutyl carbinol (MIBC). The experiment on foamability was conducted by means of a dynamic foam analyzer (DFA), producing froth by sparging the air through a porous frit, thus measuring its height at different gas delivery rates. In addition, we expanded our study by introducing particles that were oppositely charged to the bubbles and varied the zeta potential of both the particles and the bubbles. To our knowledge, such a study has never been conducted.

## 2. Materials and Methods

### 2.1. Materials

All of the chemicals and silica particles were purchased from Sigma-Aldrich (Darmstadt, Germany). The silica particles were 10 μm radii. Amino-3-methoxy silane (APTMS), ethanol and sodium hydroxide were used for the chemical modification of the silica particles. The buffers with pH = 4, pH = 7, pH = 9, and pH = 10 were used for the preparation of the solutions, by diluting with deionized water (DI) until reaching electroconductivity 92.4 μS/cm, which corresponds to ionic strength I = $7.5 \times 10^{-4}$ mol/L. DI water was produced by a water purification system (Elga Lab Water Ltd., High Wycombe, UK).

Pre-treatment with amino-3-methoxy silane (APTMS): We used the procedure described in ref. [55] to adjust the isoelectric point (IEP) of the silica particles. The surface of the particles was covered with amino groups by means of this method. Thus, the isoelectric point of the particles changed from pH ≈ 2.5 [56] to pH ≈ 9.2. This is due to a chemical reaction between amino-3-methoxy silane (APTMS) and Si–OH groups. The silica particles were positioned in 1 mol/L NaOH (T= 60 °C) and stirred. After that, they were flushed with *DI* water and soaked in a mixture of 100 mL ethanol + 2 cm$^3$ amino-3-methoxy silane (APTMS) for 24 h (T= 60 °C). Finally, they were rinsed with DI water. Figure 1 shows the chemical reaction on the surface of the silica particles. Figure 1 indicates the presence of amino groups on the chains, which were attached to the silica particles. Due to their presence, the isoelectric point was observed at pH = 9.2. We tested the foamability of 20 ppm MIBC at different pH values and constant ionic strength (I = $7.5 \times 10^{-4}$ mol/L) in the absence and presence of 1 wt.% modified -10 μm silica particles.

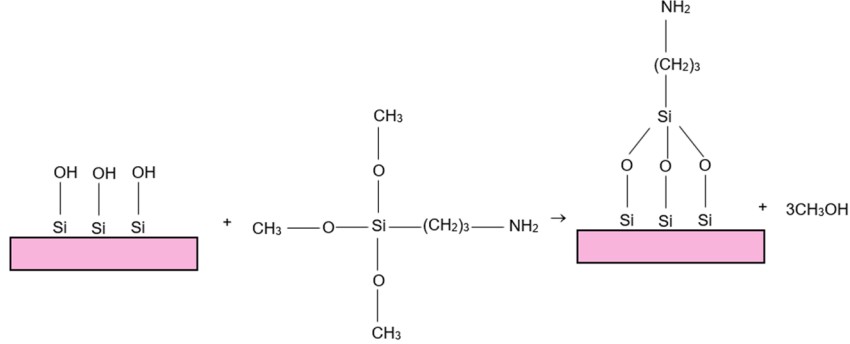

**Figure 1.** Chemical reaction on the surface of the silica particles.

### 2.2. Methods

The froth tests were conducted by means of a dynamic foam analyzer (DFA—100, Krüss Optronics GmbH, Hamburg, Germany). The froth was produced by sparging the air with a preliminary adjusted flow rate through a porous bottom in a glass column. The apparatus was controlled using a computer, with which different features of the experiment were initially set, for example, the time and flow rate of the gas delivery. The froth was

scanned with a scanline camera, which delivered the image to the computer. Hence, the height of the froth was monitored in time. We conducted the froth test with a gas delivery rate in the range of 0.2−0.5 L/min. The zeta potential of the modified silica particles at different pH values and constant ionic strength were measured by means of Zetasizer (Malvern Panalytical, Worcestershire, UK).

### 3. Results and Discussion

#### 3.1. Zeta Potential Measurements

The intrinsic isoelectric point (IEP) of the silica particles was at pH ≈ 2.5 [56]. The isoelectric point (IEP) of the modified silica particles was at pH ≈ 9.2. Figure 2 shows the zeta potential of the modified silica particles and microbubbles [57] versus pH at I = $10^{-3}$ mol/L. One can see that the microbubbles and the modified silica particles were oppositely charged ($\xi_b$ = −22 mV, $\xi_p$ = 60 mV) at pH = 5.8. The procedure did not affect the hydrophobicity of the silica particles (CA ≈ 30°).

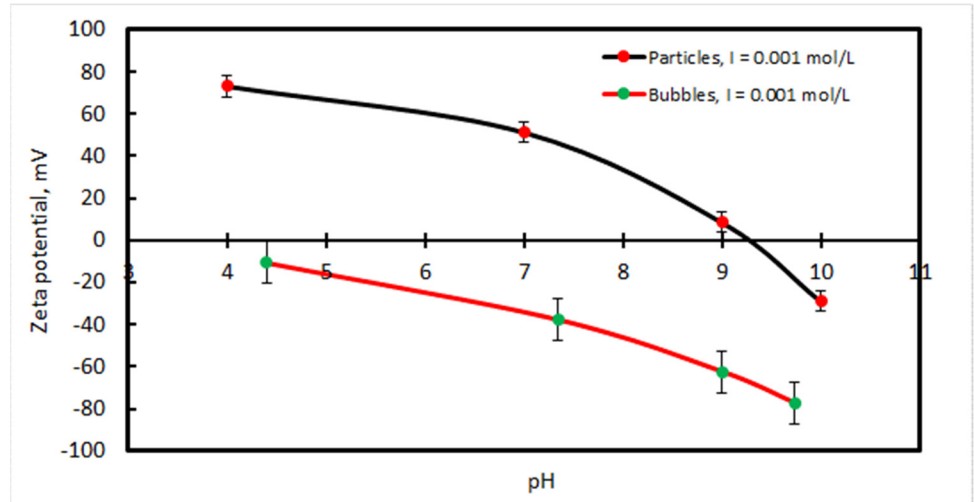

**Figure 2.** Zeta potential of modified silica particles and microbubbles [53] versus pH at I = $10^{-3}$ mol/L. The graphic of the zeta potential of bubbles is reproduced with the permission of Elsevier with license number 5372051222249.

The experiment on the foamability of the aqueous solution of 20 ppm MIBC at constant ionic strength in the absence and presence of the modified silica particles was used to analyze two basic effects: (i) the electrostatic repulsion between the bubbles; (ii) the electrostatic attraction between the bubbles and the particles.

The electrostatic disjoining pressure, assuming constant surface potential [58], can be calculated by the following formula:

$$\Pi_{el} = \frac{\varepsilon\varepsilon_0\kappa^2}{2\pi} \frac{2\Psi_{s1}\Psi_{s2}\cosh(\kappa h) - \left(\Psi_{s1}^2 + \Psi_{s2}^2\right)}{\sinh^2(\kappa h)} \tag{1}$$

where $\varepsilon$ and $\varepsilon_0$ are the static dielectric permittivities of water and free space, $F$ is the Faraday constant, $\kappa = \sqrt{2F^2 c_0/\varepsilon\varepsilon_0 RT}$ is the Debye constant (in SI unit), $c_0$ is the electrolyte concentration), $R$ and $T$ are gas constant and temperature, $\Psi_{s1}$ and $\Psi_{s2}$ are the surface potential values of the first (air/water) and the second (water/solid) surfaces, and $h$ is the thickness of the wetting film. Equation (1) is valid for the different/or opposite surface potential values of the two films' surfaces. Another well-known formula for the electrostatic

disjoining pressure, assuming the superposition approximation [59] and surface potential values with the same sign reads:

$$\Pi_{el} = 64cR_gT\tanh\left(\frac{F\Psi_{s1}}{4R_gT}\right)\tanh\left(\frac{F\Psi_{s2}}{4R_gT}\right)\exp(-\kappa h) \tag{2}$$

The van der Waals disjoining pressure, $\Pi_{vdW}$, as a function of the film thickness, $h$, can be calculated by the following formula [60]:

$$\Pi_{vdW} = -\frac{A(h,\kappa)}{6\pi h^3} + \frac{1}{12\pi h^2}\frac{dA(h,\kappa)}{dh} \tag{3}$$

where $A(h,\kappa)$ is the Hamaker–Lifshitz function, which depends on the film thickness and the Debye constant, $\kappa$, due to the electromagnetic retardation effect and is described as

$$A(h,\kappa)_{132} = \frac{3k_BT}{4}(1+2\kappa h)e^{-2\kappa h} + \frac{3\hbar\omega}{16\sqrt{2}}\frac{(n_1^2-n_3^2)(n_2^2-n_3^2)}{(n_1^2-n_2^2)}\left\{\frac{I_2(h)}{\sqrt{n_2^2+n_3^2}} - \frac{I_1(h)}{\sqrt{n_1^2+n_3^2}}\right\} \tag{4}$$

$$I_j(h) = \left[1+\left(\frac{h}{\lambda_i}\right)^q\right]^{-\frac{1}{q}} \tag{5}$$

$$\lambda_i = \frac{2\sqrt{2}c}{\omega\pi}\sqrt{\frac{1}{n_3^2(n_i^2+n_3^2)}} \tag{6}$$

where $\hbar = 1.055 \times 10^{-34}$ Js/rad is the Planck constant (divided by $2\pi$); $\omega$ is the absorption frequency in the UV region, typically around $2.068 \times 10^{16}$ rad/s for water; and $n_1$, $n_2$, and $n_3$ are the characteristic refractive indices of the two dispersion phases (air and mineral) and the medium (water). For example, $n_1^2 = 1$ for air, $n_3^2 = 1.887$ for water, and $n_2^2 = 2.359$ for crystalline quartz. Moreover, $c = 3.10^8$ m/s is the speed of light in vacuum and $q = 1.185$, $\lambda_1$ and $\lambda_2$ are characteristic wavelengths of the first (air/water) and second (water/solid) surfaces of the wetting film.

### 3.2. Foamability of 20 ppm MIBC at Different pH Values and Constant Ionic Strength $I = 7.5 \times 10^{-4}$ mol/L

As shown in Figure 2, the absolute value of the zeta potential of the bubbles increased with the increase in the pH value.

Figure 3 shows the height of the froth column of the aqueous solution of 20 ppm MIBC vs. the pH value at constant ionic strength $I = 7.5 \times 10^{-4}$ mol/L and different gas delivery rates. This ionic strength corresponded to the Debye length $1/\kappa = 11.14$ nm. This indicates that the bubbles could repel each other at approx. 33.5 nm distance from each other. Three times the Debye length practically corresponded to the thickness of the diffuse layer. One can see in the figure that the height of the froth increased with the increase in the pH value until reaching the maximum at a certain pH value in the range of pH = 8 to pH = 9. This is easy to explain by increasing the electrostatic repulsion between the bubbles—their absolute value of the zeta potential increased with the increase in the pH value.

Figure 4 presents the DLVO curves of the electrostatic, van der Waals, and the total disjoining pressure versus the distance between two bubbles at ionic strength $I = 7.5 \times 10^{-4}$ mol/L and pH = 7.34. One can see in the figure that the existence of the potential barrier hindered the ability of the bubbles to approach each other and thus coalesce. For example, the bubbles with radii 500 μm would stop thinning at about 43 nm from each other. The increase in the froth height corresponded to an increase in the foamability of the MIBC solution. The decrease in the froth height at pH > 9 was unexpected, but we suggest an explanation below.

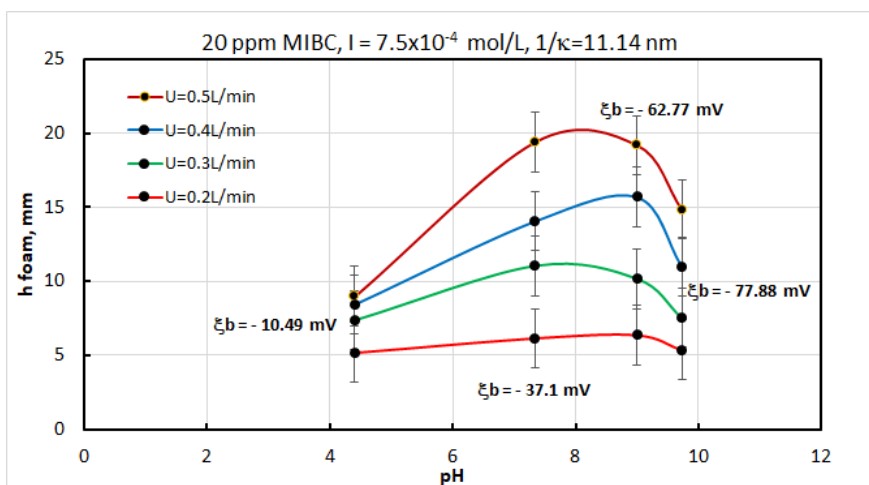

**Figure 3.** Height of the froth column of aqueous solution of 20 ppm MIBC vs. the pH value at constant ionic strength I = 7.5 × 10$^{-4}$ mol/L at different gas delivery rates; the zeta potential of the bubbles at each pH value is depicted in the figure. The error bar is ±2 mm. This behavior correlates with the DLVO theory [44,45,61].

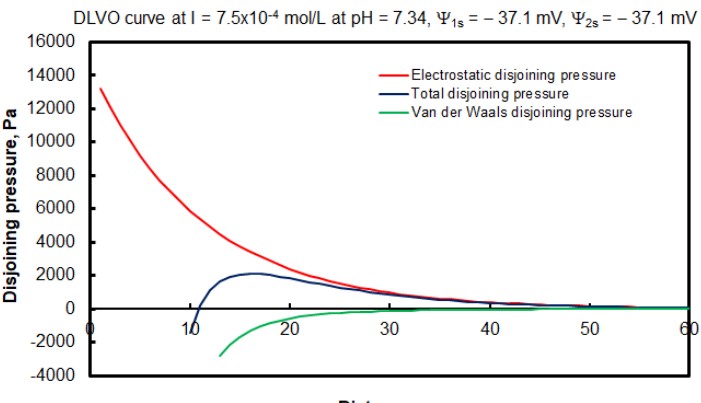

**Figure 4.** DLVO curves of the electrostatic, van der Waals, and total disjoining pressure of two approaching bubbles at ionic strength I = 7.5 × 10$^{-4}$ mol/L and pH = 7.34.

*3.3. Foamability of 20 ppm MIBC + 1 wt. % Modified Silica Particles at Different pH Values and Constant Ionic Strength I = 7.5 × 10$^{-4}$ mol/L*

Table 1 shows the values of the zeta potential of the bubbles and particles in the 20 ppm solution of MIBC. One can see the negative value of the zeta potential of the bubbles and the positive value of the zeta potential of the modified silica particles. The electrostatic attraction between the bubbles and particles would enable the bubbles to be covered with fine particles, as long as the bubbles were significantly larger than the particles. This resulted in two basic effects: (i) a decrease in the electrostatic repulsion between the bubbles and (ii) steric repulsion between the bubbles. Figure 5 shows the froth height of the 20 ppm aqueous solution of MIBC + 1 wt.% modified silica particles vs. pH and at different gas delivery rates. In Figure 5, one can see the same basic trend of reaching the maximum value of the froth height at a range of pH = 8 to pH = 9 and the decrease in the froth height at pH > 9. The froth height levels in Figure 5 were higher than the ones in Figure 3. This is expectable because of the attachment of the fine particles to the bubbles due to the electrostatic attraction. It generates additional steric repulsive force between the bubbles. Yet, a maximum froth height at a certain pH value in the range of 8 < pH < 9 can be seen. The froth height dropped at pH > 9.

**Table 1.** Values of pH and the corresponding zeta potentials of the bubbles and the modified silica particles in suspension at $7.5 \times 10^{-4}$ mol/L ionic strength.

| pH | 4.40 | 7.42 | 8.88 | 9.40 |
|---|---|---|---|---|
| **Zeta potential bubbles, mV** | $-10.49$ | $-36.93$ | $-57.82$ | $-66.56$ |
| **Zeta potential particles, mV** | 72.25 | 45.55 | 12.43 | $-4.84$ |

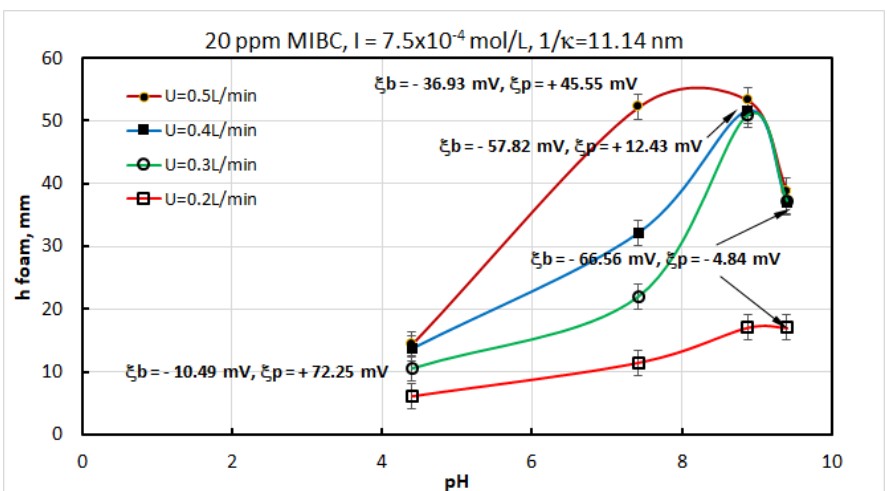

**Figure 5.** Height of the froth column of aqueous solution of 20 ppm MIBC + 1 wt.% modified fine silica particles vs. the pH value at constant ionic strength I = $7.5 \times 10^{-4}$ mol/L at different gas delivery rates; the zeta potential of the bubbles at each pH value is depicted in the figure. The error bar is ±2 mm.

Figure 6 shows the DLVO curves of the electrostatic, van der Waals, and the total disjoining pressure vs. distance between the bubbles and the modified silica particles at ionic strength I = $7.5 \times 10^{-4}$ mol/L and at pH = 4.4. One can see that the total disjoining pressure became negative at a distance below 20 nm.

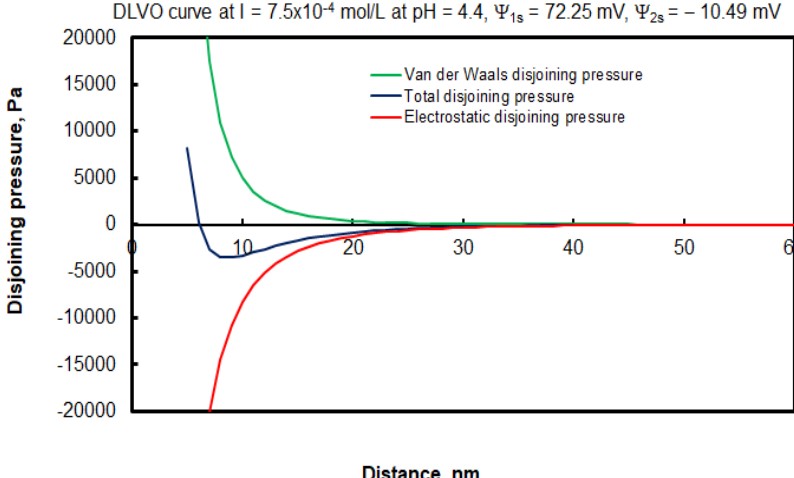

**Figure 6.** DLVO curves of the electrostatic, van der Waals, and total disjoining pressure of bubble approaching silica particle at ionic strength I = $7.5 \times 10^{-4}$ mol/L.

The existence of a maximum froth height at 8 < pH < 9 and the drop in the froth height at pH > 9 correlated with the solubility of carbon dioxide, which reached its minimum at pH = 8.36 [62–64]. Water at pH > 8.36 was free of dissolved carbon dioxide. Hence, at pH > 8.36, the water became super-saturated with carbon dioxide. This is an excellent

condition for gas precipitation during froth generation. We could not find any studies on the effect of the precipitation of gas on dispersed systems, but such an effect exists. For example, an increase in gas concentration breaks the emulsion films, while the films are durable at decreased gas concentrations [65]. Surfactant-free emulsions can become very durable if they are degassed [66]. These works do not report on the precipitation of gas but only on the effect of gas concentration in an aqueous medium on the stability of thin emulsion films and emulsion. Research that addresses this effect on thin foam films and froths is lacking in the literature.

### 4. Conclusions

We arrived at the following conclusions with this work:

1. The foamability of 20 ppm aqueous solutions of methyl isobutyl carbinol (MIBC) depended on the pH value and consequently the electrostatic repulsion between the bubbles at ionic strength I = $7.5 \times 10^{-4}$ mol/L ($1/\kappa$ = 11.14 nm) until reaching a maximum between 8 < pH < 9. The increase in the electrostatic repulsion between the bubbles prevented their coalescence and hence boosted the foamability.

2. The electrostatic attraction between the bubbles and 1 wt. % modified silica particles increased the froth height, compared with the particle-free sample, until reaching its maximum at 8 < pH < 9. This was due to the steric repulsion between the bubbles. The latter additionally prevented the coalescence between the bubbles.

3. The existence of the maximum froth height at 8 < pH < 9 can be explained by the lack of the solubility of carbon dioxide at pH = 8.36. Hence, at pH > 8.36, the water was free of dissolved carbon dioxide. The solution under such conditions was pre-saturated with carbon dioxide. This is an excellent condition for gas precipitation during froth generation. The gas precipitation of the dissolved gas contributes to the coalescence of the bubbles. This is likely the reason for the drop in the froth height at pH > 8.36. This explanation remains a hypothesis until it can be either proven or rejected.

The next steps of our studies are to vary the strength of the electrostatic repulsions by performing experiments with different constant ionic strength values.

**Author Contributions:** Experimental methodology, O.O.; validation, S.I.K. and N.A.G.; investigation, S.I.K., N.A.G., K.M. and S.H.; resources S.I.K., N.A.G. and S.H.; writing—review and editing, S.I.K.; supervision, O.O. and S.I.K.; project administration, S.I.K. All authors have read and agreed to the published version of the manuscript.

**Funding:** This paper is supported by European Union's Horizon 2020 research and innovation program under Grant Agreement No. 821265, project FineFuture (Innovative technologies and concepts for fine particle flotation: unlocking future fine-grained deposits and Critical Raw Materials resources for the EU).

**Institutional Review Board Statement:** Not applicable.

**Informed Consent Statement:** Not applicable.

**Data Availability Statement:** The authors confirm that the data supporting the findings of this study are available within the article.

**Conflicts of Interest:** The authors declare no conflict of interest.

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
