# Peer review of "Electrostatic Forces in Control of the Foamability of Nonionic Surfactant"

_coatings, doi:10.3390/coatings13010037_

Round 1
Reviewer 1 Report
The comments are on the attached file.

Author Response
Dear Reviewer 1,
We would like to thank you for the review. We hope that you will be satisfied with the new version of the manuscript and with the response to your comments.
Please find bellow the response to your comments.
Best regards,
Stoyan Karakashev
coatings-2129207: Stoyan I. Karakashev, Nikolay A. Grozev, Svetlana Hristova, Kristina Mircheva, Orhan Ozdemir, “Electrostatic Forces in Control of the Foamability of Nonionic Surfactant”
The authors have performed a study of the foamability of methyl isobutyl carbinol. The research topic is of high scientific interest. However, the manuscript is in need of some improvement. Some suggestions and a mandatory requirement are indicated below.
- Page 1 – Line 24
Change “explain” to “explained”
- Page 1 – Line 27
Change “remains hypothesis” to “remains as a hypothesis”
- Page 1 – Line 31
Change “producing” to “production”
- Page 2 – Lines 58-59
Change “To our knowledge such a study has never been conducted in the literature.” to “To our knowledge such a study has never been conducted.”
- Page 2 – Line 82
Change “pH value” to “pH values”
- Page 5 – Line 162
Change “For examples” to “For example”
Response: Thank you. We made the corrections.
- Page 7 – Figure 7
Figure 7 is not mentioned nor described in the text of the manuscript, and it is not an original figure of the authors for which a mandatory reproduction permission is required.
The shown figure 7 was published originally in the book CO2 in Seawater: Equilibrium, Kinetics, Isotopes
- Wolf-Gladrow and R.E. Zeebe, Elsevier Science (2001) from where a reproduction permission is needed.
An extreme alternative is not to show figure 7, describe it and cite the original source, which is not the review article. For this I suggest to attain a copy of the book, Paperback ISBN: 9780444509468 and review it first.
Additional comments:
The caption of Figure 7 in the submitted manuscript is: “Figure 7. Fraction of CO2, H2CO3, HCO3- and CO32- versus pH [58]”. Reference 58 is a review paper for which a reproduction permission per figure that were not of the authors was given.
Response: We removed Fig.7 as far as we could get license, but just cited literature sources about the lack of solubility of CO2 at pH > 8.3.

Reviewer 2 Report
The manuscript by Karakashev and co-workers reports the heights of the froth column of aqueous solutions of methyl isobutyl carbinol in the absence and presence of amino-modified 10E-5 m silica particles at various pH values and gas flow rates. In the presence of the particles, the heights of the froth column increase up to 5 times. The heights are maximum at a pH in the range of 8-9. These changes are explained within the frameworks of the DLVO theory. In contrast to this theory, the heights of the froth column decrease at a pH greater than 9. The authors attribute this effect to the solubility of carbon dioxide.
The topic of the study is relevant in the field and its motivation is justified. Cited references relate to the research topic and are justified. The objectives, experimental methods, and main findings of the study are clearly presented. However, the conclusions drawn need to be corrected.
The hypothesis about the effect of carbon dioxide solubility on foam formation seems far-fetched. The three major constituents of air are nitrogen, oxygen, and argon. The mole fraction of CO2 is only 0.03% (volume). The effect may appear after the pH of the solution has changed, but should be gone once equilibrium is reached.
Lines 49-51: “The intrinsic negative surface potential of the bubbles in water at normal pH value is still not well understood, but it is proved that it depends on the pH value [49]. Their isoelectric point is at pH ≈ 4 [1, 50].”
Do you mean a pure aqueous solution of methyl isobutyl carbinol? At what concentration? Why then is the Zeta potential at pH 4.4 is -10.49 mV?
Lines 134-135: “Fig. 1 shows the van der Waals, electrostatic and the total disjoining pressures versus distance between the bubble and the particle.”
Fig. 1 shows “Reaction of amino – 3 – methoxy silane (APTMS) with the silica surface Si−OH groups.”
Table 1 is not needed as the required values are given in Table 2.
Author Response
Dear Reviewer 2,
We would like to thank you for the review. We hope that you will be satisfied with the new version of the manuscript and with the response to your comments.
Please find bellow the response to your comments.
Best regards,
Stoyan Karakashev
The manuscript by Karakashev and co-workers reports the heights of the froth column of aqueous solutions of methyl isobutyl carbinol in the absence and presence of amino-modified 10E-5 m silica particles at various pH values and gas flow rates. In the presence of the particles, the heights of the froth column increase up to 5 times. The heights are maximum at a pH in the range of 8-9. These changes are explained within the frameworks of the DLVO theory. In contrast to this theory, the heights of the froth column decrease at a pH greater than 9. The authors attribute this effect to the solubility of carbon dioxide.
The topic of the study is relevant in the field and its motivation is justified. Cited references relate to the research topic and are justified. The objectives, experimental methods, and main findings of the study are clearly presented. However, the conclusions drawn need to be corrected.
The hypothesis about the effect of carbon dioxide solubility on foam formation seems far-fetched. The three major constituents of air are nitrogen, oxygen, and argon. The mole fraction of CO2 is only 0.03% (volume). The effect may appear after the pH of the solution has changed, but should be gone once equilibrium is reached.
Response: We completely agree with the reviewer. This hypothesis needs strong proof. Such one could be equilibrated solution at pH = 9 and we intend to check this in near future not only for bubble – bubble interaction but with bubble – hydrophobic particles interaction. Yet, we must remind here that we suggest only hypothesis here and we don’t insist that this hypothesis is true, but it is just opinion. I hope the reviewer doesn’t mind if we leave this hypothesis as an object if thinking of the readers.
Lines 49-51: “The intrinsic negative surface potential of the bubbles in water at normal pH value is still not well understood, but it is proved that it depends on the pH value [49]. Their isoelectric point is at pH ≈ 4 [1, 50].” Do you mean a pure aqueous solution of methyl isobutyl carbinol? At what concentration?
Response: The presence of MIBC does not affect the zeta potential of the bubbles because it is nonionic frother. So, this dependence is valid for both pure water and aqueous solution of MIBC. In this particular case we have 20 ppm (2x10-4 mol/L) MIBC.
Why then is the Zeta potential at pH 4.4 is -10.49 mV?
Response: We used the Gouy – Chapmen theory to calculate the zeta potential at I = 7.5x10-4 mol/L ionic strength based on the experimental data of zeta potential of bubbles at I = 0.01 nol/L published in ref. [1]
Lines 134-135: “Fig. 1 shows the van der Waals, electrostatic and the total disjoining pressures versus distance between the bubble and the particle.”
Fig. 1 shows “Reaction of amino – 3 – methoxy silane (APTMS) with the silica surface Si−OH groups.”
Response: Thanks a lot for letting us know about this error. We corrected it.
Table 1 is not needed as the required values are given in Table 2.
Response: We agree and consequently we removed Table 1 and made the appropriate corrections in the text.
Literature.
- Yang, C.; Dabros, T.; Li, D.; Czarnecki, J.; Masliyah, J. H., Measurement of the zeta potential of gas bubbles in aqueous solutions by microelectrophoresis method. J. Colloid Interface Sci. 2001, 243, (1), 128-135.

Reviewer 3 Report
Manuscript titled, “Electrostatic forces in control of the foamability of nonionic surfactant by Karakashev et al. deals with the study of foamability of 20 ppm methyl isobutyl carbinol at constant ionic strength I = 7.5×10-4 mol/L at different pH value in absence and presence of modified silica particles. Authors have established an increase of the foamability with the increase of the pH value until pH = 8.3, beyond which it decreases. The data are of good quality. Therefore, my suggestion is that the manuscript can be accepted by Coatings after minor revisions. The suggestions are shown as follows:
1. Some minor mistakes need to be revised. Line 24, replace “explain” with explained; Line 70, “detailс”; etc.
2. I suggest that the authors should add some recent references in the introduction section of the manuscript.
3. It is strongly recommended to provide reference for line 165 for its detailed explanation.
4. The source of deionized water utilized for the dilution of solutions should be added in text.
5. Conclusion section of the manuscript can be improved.
6. References should be revised:
a) Year should be written in bold.
b) Appropriate abbreviations should be used for the journals for e.g. “Journal of Physical Chemistry C” as J. Phys. Chem. C; “Journal of Colloid and Interface Science” as J. Colloid Interface Sci.; “Current Opinion in Colloid and Interface Science” as Curr. Opin. Colloid Interface Sci. etc.

Author Response
Dear Reviewer 3,
We would like to thank you for assessing our manuscript. I took into consideration your comments and suggestions and prepared a new draft of the manuscript. We hope you will be satisfied with the new revised version and with the response to your comments and questions. Please find them here bellow.
With best regards,
Stoyan Karakashev
Manuscript titled, “Electrostatic forces in control of the foamability of nonionic surfactant by Karakashev et al. deals with the study of foamability of 20 ppm methyl isobutyl carbinol at constant ionic strength I = 7.5×10-4 mol/L at different pH value in absence and presence of modified silica particles. Authors have established an increase of the foamability with the increase of the pH value until pH = 8.3, beyond which it decreases. The data are of good quality. Therefore, my suggestion is that the manuscript can be accepted by Coatings after minor revisions. The suggestions are shown as follows:
- Some minor mistakes need to be revised. Line 24, replace “explain” with explained; Line 70, “detailс”; etc.
Response: Corrected.
- I suggest that the authors should add some recent references in the introduction section of the manuscript.
Response: We added 5 recent references.
- It is strongly recommended to provide reference for line 165 for its detailed explanation.
Response: We added references.
- The source of deionized water utilized for the dilution of solutions should be added in text.
Response: We added it to the text.
- Conclusion section of the manuscript can be improved.
Response: We added more elucidations in the conclusion section.
- References should be revised:
- a)Year should be written in bold.
- b)Appropriate abbreviations should be used for the journals for e.g. “Journal of Physical Chemistry C” as J. Phys. Chem. C; “Journal of Colloid and Interface Science” as J. Colloid Interface Sci.; “Current Opinion in Colloid and Interface Science” as Curr. Opin. Colloid Interface Sci. etc.
Response: We revised it.

Round 2
Reviewer 1 Report
The comments are on the attached file.

Author Response
Dear Reviewer 1,
We would like to thank you for assessing our manuscript. We hope you will be satisfied with the new revised version and with the response to your comments and questions. Please find them here bellow.
With best regards,
Stoyan Karakashev
The authors have presented an improved version of the manuscript. Nevertheless, a new minor correction is needed:
- Page 1 – Line 59
Change “conducted..” to “conducted.” (There is an extra period.)
Response: We did it.
